# A Novel Antibiotic, Rhodomyrtone: Pharmacokinetic Studies in a Murine Model and Optimization and Validation of High-Performance Liquid Chromatographic Method for Plasma Analysis

**DOI:** 10.3390/antibiotics13020156

**Published:** 2024-02-05

**Authors:** Tan Suwandecha, Boon-Ek Yingyongnarongkul, Kanokkan Towtawin, Supayang Piyawan Voravuthikunchai, Somchai Sriwiriyajan

**Affiliations:** 1School of Pharmacy and Drug and Cosmetic Excellence Center, Walailak University, Thaiburi, Thasala District, Nakhon Si Thammarat 80160, Thailand; tan.su@wu.ac.th; 2Department of Chemistry and Center of Excellence for Innovation in Chemistry (PERCH-CIC), Faculty of Science, Ramkhamhaeng University, Bangkok 10240, Thailand; boonek@ru.ac.th; 3Division of Health and Applied Science, Faculty of Science, Prince of Songkla University, Hat Yai, Songkhla 90110, Thailand; kanokkan.towtawin@gmail.com; 4Center of Antimicrobial Biomaterial Innovation-Southeast Asia and Natural Product Research Center of Excellence, Faculty of Science, Prince of Songkla University, Hat Yai, Songkhla 90110, Thailand; supayang.v@psu.ac.th

**Keywords:** rhodomyrtone, antibiotics, pharmacokinetics, HPLC, method validation

## Abstract

Rhodomyrtone has indisputable and undeniable potential as a new antibiotic for antibiotic-resistant Gram-positive bacteria. Therefore, the main objective of this study was to determine the pharmacokinetics profiles of orally administered rhodomyrtone in rats. A reverse-phase HPLC-UV method was developed, optimized and validated for the analysis of rhodomyrtone concentrations in rat plasma. The retention time of papaverine and rhodomyrtone was 3.928 and 5.937 min, with no interference with the excipients used. The lower limit of quantification (LLOQ) of rhodomyrtone in the plasma sample was 0.04 μg/mL, the accuracy of rhodomyrtone at the LLOQ level ranged from 93.64 to 106.36%, precision was 6.59%, 80–120% for accuracy and <20% CV for precision. The calibration curve was linear at concentrations ranging from 0.04 to 128 µg/mL with a correlation coefficient (r) value of equal to or greater than 0.999. Sprague Dawley rats received a single dose of rhodomyrtone at 50 and 100 mg/kg. Blood samples were collected from tail veins. The peak plasma concentration was observed at 2 h, and the area under the curve of rhodomyrtone at 50 mg/kg and 100 mg/kg was 3.41 ± 1.04 and 7.82 ± 1.53 μg·h/mL, respectively. The results demonstrated linear pharmacokinetics characteristics at the studied dosage range. The plasma concentration of rhodomyrtone was above the minimal inhibition concentrations of several common pathogenic bacteria of medical importance. The proposed HPLC-UV method is fast, cost-effective, reliable and reproducible, and it is proposed for the routine analysis of rhodomyrtone.

## 1. Introduction

Rhodomyrtone, an acylphloroglucinol natural antibiotic with a molecular weight of 442.5 g/mol, was originally isolated from the leaves of *Rhodomyrtus tomentosa* (Aiton) Hassk [1,2], displaying a pronounced activity against Gram-positive bacteria. In the search for a novel antibiotic with a new mode of action, the very distinct antibacterial potency of rhodomyrtone attracted our research group to work on this target molecule. The compound presented extremely strong and broad-spectrum activity against *Staphylococcus aureus* [3,4,5,6], *Enterococcus faecalis* [7], *Streptococcus pyogenes* [8], *Strptococcus mutans* [9], *Cutibacterium acnes* [9,10], capsule-forming *Streptococcus pneumoniae* [11], *Bacillus cereus* [12] and endospore-producing *Clostridium difficile* [13]. Of interest, this compound was found to be highly potent against problematic multidrug-resistant isolates, including methicillin-resistant *S. aureus* (MRSA) [14,15], vancomycin-intermediate *S. aureus* (VISA) and vancomycin-resistant enterococci (VRE) [16], with the MIC values at concentrations comparable with or superior to last-resort antibiotics in the glycopeptide group. In addition, its ability to inhibit biofilm forming and killing pathogens within established biofilms was demonstrated in staphylococci [17,18] and *C. acnes* [19]. More importantly, after 45 passages of stepwise isolation of *S. aureus* or MRSA on Luria-Bertani agar supplemented with the natural antibiotic rhodomyrtone, the bacterial cells were not found to be resistant to rhodomyrtone [20].

Multidisciplinary research efforts were made by several international collaborative research groups to gain comprehensive information on rhodomyrtone in the domain of drug discovery. The alleged target sites of rhodomyrtone on MRSA were on the cytoplasmic membrane and cell wall by analyzing proteomic and transcriptomic profiles [20,21,22]. Studies on the mode of action disclosed that the main target of rhodomyrtone was on the bacterial cell membrane, which is vastly different from any conventional antibiotic targets, such as cell wall biosynthesis, DNA-related enzyme inhibition, translation or transcription [23,24]. Many membrane-active antimicrobial peptides are typical membrane-inserting molecules [25]. On the other hand, being uncharged and devoid of a particular amphipathic membrane-interacting domain, the bacterial cytoplasmic membrane is disturbed by rhodomyrtone through a distinct molecular mechanism that leads to inward folding of the cell membrane and internalization of plasma membrane proteins [24]. Membrane-bound transport proteins are a crucial antibiotic resistance mechanism that render many common antibiotics ineffective. Rhodomyrtone has a unique mechanism that inspires new treatment approaches to fight against antibiotic-resistant bacteria. By trapping proteins in membrane vesicles, rhodomyrtone inhibits the bacterial membrane function, which involves a multitude of cellular processes that are vital for cell survival. This could be a new strategy to inactivate multi-drug efflux pumps. In addition, it enhances bacterial cell membrane uptake, which may potentiate the effectiveness of other antibacterial agents.

The superior activity of rhodomyrtone over vancomycin and its new mode of action warrant further studies on the pharmacokinetics of this new molecule. Therefore, this study aimed to (i) develop and validate an economical, fast, simple, accurate and precise chromatographic method for the analysis of rhodomyrtone concentrations in plasma and (ii) establish pharmacokinetic studies on this novel antibiotic. 

## 2. Results and Discussion

### 2.1. Method Validation

The HPLC method was used to quantify the concentration of rhodomyrtone in rat plasma after a single oral administration of 50 and 100 mg/kg. Blood samples were collected from tail veins at 0, 0.25, 0.5, 1, 2, 3, 4, 5, 6, 7 and 8 h. Rat plasma samples were prepared as described in Section 3.3. Rhodomyrtone was analyzed using a Bondapak C18 HPLC column at 25 °C. The maximum injection volume was found to be a volume of 80 μL at a flow rate of 1 mL/min. After several trials, the mobile phase was optimized to achieve good resolution and symmetric peak shapes for both rhodomyrtone and papaverine. The optimum mobile phase consisted of a mixture of acetonitrile/water (80:20), %*v*/*v*, which resulted in good separation and resolution. An appropriate wavelength at 302 nm gave good sensitivity, with no interfering substance of other components. The method was validated and showed an acceptable calibration curve, precision, accuracy, selectivity, specificity, recovery and stability. The curve of the rhodomyrtone concentration–time profiles was fitted in a non-compartment pharmacokinetic model due to the whole organism as a single compartment where the drug distributes homogeneously and instantaneously, resulting in a constant volume of distribution. The non-compartment model is multifaceted, and it is acceptable for bioequivalence studies.

The specificity of the assay was assessed by analyzing blank plasma spiked with rhodomyrtone at a concentration of 16 µg/mL. Chromatograms illustrated a good separation between rhodomyrtone and IS. The chromatogram of blank rat plasma showed no interference peak (Figure 1A). There were no endogenous peaks observed at the retention time of rhodomyrtone (5.937 min) and IS (3.928 min) (Figure 1B). The lower limit of quantification (LLOQ) of rhodomyrtone in the plasma sample was 0.04 μg/mL, indicating high sensitivity of the assay. The accuracy of rhodomyrtone at the LLOQ level ranged from 95.14 to 113.83% (Table 1) and precision was 6.59%. The results were in the acceptable range of 80–120% for accuracy and <20% CV for precision. The calibration curve was linear at concentrations ranging from 0.04 to 128 µg/mL, with a correlation coefficient (r) value of equal to or greater than 0.999. The main advantage of this method is its sensitivity and simplicity, which makes it suitable for both specific pharmacokinetic studies as well as routine laboratory assays. In addition, a simple protein precipitation step, the low cost and short run time encourage future operation of this assay.

The assay’s precision and accuracy were evaluated through the examination of five replicates, each containing quality control (QC) samples of rhodomyrtone at three different concentrations (0.1, 50 and 100 µg/mL). The intra-day demonstrated % accuracy between 94 and 108% with less than 12% CV. The inter-day results showed relative standard deviation (RSD) ranging between 95 and 106%, with less than 10% CV (Table 2). Both intra-day and inter-day met the accuracy (%) and precision (% CV) acceptance criteria for each QC (85–115%) and ±15% CV. Analytical recovery was determined by calculating the peak area ratios of rhodomyrtone and IS of pre-extraction spiked samples versus samples spiked after extraction at the concentration of QC samples. The percentage recovery of rhodomyrtone at the concentrations 0.1, 50 and 100 µg/mL was 102.36 ± 3.36, 100.83 ± 2.08 and 100.68 ± 1.96, respectively. Long-term stability evaluation was performed to demonstrate the stability of the analyte in the matrix for a longer duration. The accuracy of rhodomyrtone was 94–114%, and the precision was <13.5%. The bench-top stability evaluation revealed that the accuracy was 92–109% and the precision was <10% as shown in Table 3. The accuracy and precision after the freeze–thaw cycles were 106–110% and <5%, respectively (Table 4). 

### 2.2. Pharmacokinetics Study of Rhodomyrtone

Pharmacokinetic studies of rhodomyrtone have not yet been reported. Many drugs failed to demonstrate clinical effects due to the lack of an appropriate pharmacokinetic profile [26]. The drug concentration in the blood circulation is crucial for drug effectiveness, and an inadequate concentration will result in treatment failure. The antimicrobial concentration below the minimal inhibition concentration (MIC) may hinder the drug effectiveness and lead to rapidly developing resistance [27]. Treated animals received a single oral administration of rhodomyrtone at concentrations of 50 mg/kg and 100 mg/kg. The mean plasma concentration–time of rhodomyrtone after rhodomyrtone administration is illustrated in Figure 2. The rhodomyrtone concentration at 8 h for doses of 50 mg/kg rhodomyrtone was about 0.1 μg/mL which is above the LLOQ. Thus, the LLOQ of the HPLC method is adequate for the quantitation of rhodomyrtone in rat plasma in this study.

Key pharmacokinetic parameters can be generated from PK profiles following oral dosing. Details on pharmacokinetic parameters that are important in therapeutic drug monitoring are shown in Table 5. 

The concentration–time profile oral administration and the time to maximum plasma concentration (Tmax) are approximately 2 h with both 50 and 100 mg/kg doses, indicating that rhodomyrtone had a reasonable rate of absorption. Rhodomyrtone 50 mg/kg up to 100 mg/kg resulted in linear pharmacokinetics behavior. Two anti-MRSA antibiotics, vancomycin and linezolid, were chosen to compare with rhodomyrtone. Vancomycin is a parenteral drug of choice [28], while linezolid is an orally administered option for invasive MRSA infection [29]. The absorption rate of rhodomyrtone was significantly slower than linezolid, which has a Tmax in rats between 10 and 30 min of oral dosing [30]. The Tmax of vancomycin is not available due to being practically not absorbed via the oral route [31]. However, oral vancomycin is recommended for treating *C. difficile* infection [32] and rhodomyrtone also has potential [13]. The SwissADME [33] model predicted that rhodomyrtone had high lipophilicity, poor water solubility and good gastrointestinal absorption (Table 6). Thus, we hypothesized that rhodomyrtone is a biopharmaceutics classification system (BCS) class II compound [34]. However, absolute bioavailability, membrane permeation, pH solubility profile and related physicochemical properties studies are needed to confirm this assumption [35]. Plasma rhodomyrtone concentrations at 30 and 15 min after oral administration of both doses were above the MIC of common pathogens (Table 7), such as MRSA, *Staphylococcus epidermidis* and *Bacillus cereus* [36]. It was approximately 4 h for doses of 50 mg/kg and 8 h for 100 mg/kg for rhodomyrtone concentrations above MIC (0.5–1 μg/mL) of most staphylococcal isolates. The evidence indicated that the dosages used in this study can be applied to control infections. There was no sign of entero-hepatic recirculation after the oral dose within 8 h of the experiment. However, more extended blood samplings and charcoal administration are needed for further confirmation [37]. 

The single oral dose of 50 and 100 mg/kg body weight produced maximum plasma rhodomyrtone concentrations (Cmax) at 0.57 and 1.31 μg/mL, respectively, indicating that the peak serum concentration that rhodomyrtone achieved in the body was in a dose-dependent manner. The Cmax of both doses was above the MBC values for pathogens of medical importance, such as *Staphylococcus aureus*, MRSA and *Bacillus cereus*. Interestingly, the dose at 100 mg/kg resulted in a Cmax of approximately 2–3-times the MBCs for MRSA (0.03–1.00 μg/mL) [36]. This is an essential piece of early evidence, showing that rhodomyrtone is a promising compound for treating serious infections caused by antibiotic-resistant bacteria. Additionally, the primary advantage of rhodomyrtone for systemic MRSA infection treatment is that it is orally active with a dose range from 50 to 100 mg/kg. SwissADME bioavailability score and AUC showed that rhodomyrtone probably has around 10% oral bioavailability, while gastrointestinal absorption is high (Table 6). The assumptions for poor oral bioavailability are poor water solubility and hepatic first-pass metabolism of rhodomyrtone. According to EMA Guidance on the Investigation of Bioavailability and Bioequivalence (CPMP/EWP/QWP/1401/98), the AUC_0–8_ covered about 80% of AUC_0–∞_ indicating that the 8 h sampling time was adequate for pharmacokinetics studies [40]. The dose-normalized AUC of rhodomyrtone between 50 and 100 mg/kg doses was not different. This implies no saturation in the absorption processes in the gastrointestinal tract. Oral rhodomyrtone has inferior bioavailability when compared with linezolid. Orally administered linezolid (50 mg/kg) in rats could achieve a Cmax of 25 mg/L [41]. Vancomycin, on the other hand, has an oral bioavailability below 0.5% and 10% in rats and humans, respectively [42,43]. This study is the first pharmacokinetics report on rhodomyrtone, so we decided to use the oral route to study the drug pharmacokinetics. In order to obtain the absolute bioavailability of rhodomyrtone, intravenous pharmacokinetics data are yet to be obtained. As the compound is rather insoluble, a preparation as a biocompatible solution for IV injection is required. Pharmacokinetic parameters of rhodomyrtone showed that the AUCs of 50 and 100 mg/kg doses were 3.41 ± 1.04 and 7.82 ± 1.53 μg·h/mL, respectively. The dose-normalized AUC and Cmax were not statistically different (*p* > 0.05), indicating that the extent of absorption is directly proportional to the administered doses. The apparent total volume of distribution (Vz/F) was approximately 50 L/kg and was not significantly different between 50 and 100 mg/kg doses. It indicated that drug distribution was unsaturated at the single dose of 50 to 100 mg/kg. Oral clearance (CL/F) was almost the same for the doses of 50 and 100 mg/kg, implying that rhodomyrtone follows first-order kinetics because the clearance of a compound should be a constant for first-order rate processes [44]. The terminal elimination phase half-life of rhodomyrtone (t1/2) was approximately 2.5 h. Surprisingly, the half-life of rhodomyrtone in this study was more prolonged than the half-life of intramuscular and intraperitoneal administration of vancomycin (0.6 h) [45,46] and oral linezolid (1.1 h) [47]. This suggested that rhodomyrtone could have a longer dosing interval to eradicate infectious organisms than that occurs with vancomycin. It is generally known that the half-life of a drug in rats is shorter than in humans due to differences in the metabolic rate and elimination of drugs between the two species. Rats tend to have a faster metabolism and drug elimination than humans [48]. The constant clearance and half-life, independent of doses, also confirmed the linear pharmacokinetics behavior of rhodomyrtone [49]. It is well documented that the pharmacokinetics parameters from animal studies can be used to hypothesize the prediction of human response to a drug. In the present study, the linear pharmacokinetics characteristics of rhodomyrtone were validated on the doses of 50 to 100 mg/kg. The high dose corresponds to 1/20 LD50 (2000 mg/kg) [50]. The results from this study will help in predicting the dose regimen for human clinical trials.

## 3. Materials and Methods

### 3.1. Reagents and Equipment

Rhodomyrtone (Figure 3A), an acylphloroglucinol, was isolated from *R. tomentosa*. Briefly, dried ground leaves were extracted with ethanol. Column chromatography with silica gel as a stationary phase and hexane and hexane-ethyl acetate eluents were used for the purification process of the crude extract. Nuclear magnetic resonance, infrared spectroscopy and mass spectral data were used to confirm the purity of rhodomyrtone [51]. An internal standard (IS), papaverine (Figure 1B), and dimethyl sulfoxide (DMSO) were supplied by Sigma Chemical Co. (St. Louis, MO, USA). Acetonitile was HPLC grade and supplied by Fisher Scientific (Pittsburgh, PA, USA). 

### 3.2. Preparation of Rhodomyrtone and Reference Standard Solutions

Rhodomyrtone stock solution was prepared by dissolving 10 mg of rhodomyrtone in 1 mL DMSO; then, we adjusted volume to 10 mL with acetonitrile. Papaverine stock solution was prepared by dissolving 10 mg of papaverine in 10 mL acetonitrile. The final concentration of stock solutions was 1 mg/mL of rhodomyrtone and 1 mg/mL of papaverine. All stock solutions were stored at a temperature of −20 °C until further use.

The working standard solution of rhodomyrtone was prepared by pipetting the 256 µL of rhodomyrtone stock solution and diluting with blank plasma to 1 mL. The internal standard working solution was prepared by pipette 100 µL of papaverine stock solution and diluted with acetonitrile to 1 mL. The final concentration of the working standard solution was a concentration of 256 µg/mL and papaverine 100 µg/mL.

The working standard solution of rhodomyrtone was further diluted to an appropriate concentration range for linearity, selectivity, sensitivity, accuracy and precision validation. The validation samples were prepared using the method in Section 3.3.

### 3.3. Sample Preparation

An aliquot of each rat plasma sample (100 µL) or validation sample was pipetted into a 1.5 mL microcentrifuge tube, spiked with 25 µL papaverine working standard, then the rat plasma was deproteinized with 75 µL acetonitrile, mixed for 30 s and dropped off 5 min to precipitate the protein. The insoluble content was separated from plasma by centrifugation at 5000× *g* at 10 °C for 15 min, and we collected the supernatant. All samples were filtered through a 0.22 μm PTFE membrane filter. Subsequently, each sample was transferred into glass insert vials. The HPLC injection volume was 80 µL.

### 3.4. HPLC Conditions

HPLC system consisted of Waters Alliance 2695, a model 2487 UV-Vis detector (Waters Associates, Milford, MA, USA). The maximum UV absorbance of rhodomyrtone was detected from 250 to 310 nm. Therefore, the detection wavelength was set at wavelength 302 nm. The chromatographic data were collected and analyzed using Millenium 32^®^ software. Waters Bondapak C18 column, 5 μm, 4.6 mm × 150 mm, was used to set the standard for HPLC drug analytical measurement procedure. The mobile phase consisted of a mixture of acetonitrile and deionized water (80:20% *v*/*v*). The mobile phase was pre-filtered through a 0.20 μm filter and was delivered at a rate of 1 mL/min. 

### 3.5. Validation of HPLC Method

The analytical procedure was validated according to International Conference on Harmonization (ICH) Q2R2 guidelines on validation of analytical procedures: Text and methodology, current step 4 version, November 2005. The guidelines covered 4 domains for validation of analytics, including selectivity/sensitivity, linearity range, accuracy/precision and robustness [52].

In this study, we validated the HPLC method, covering linearity, selectivity, sensitivity, accuracy, precision and sample storage condition. Standard rhodomyrtone spiked in plasma was used to assess the interference of endogenous compounds. The lowest concentration that produced an accuracy (defined as the percentage of the measured concentration to the theoretical concentration) of between 80 and 120% of the theoretical value, and a precision of less than 20% RSD, was designated as the LLOQ. The quantitation limit is generally determined by the analysis of samples with known concentrations of analyte and by establishing the minimum level at which the analyte can be quantified with acceptable accuracy and precision.

The linearity test was performed in the range of 0.04–128.0 μg/mL of rhodomyrtone in plasma samples. The least-squares linear regression line (without a weighting component) yielded the peak area ratios of each rhodomyrtone curve, which were then shown together with their correlation coefficients.

The inter- and intra-assay precision and accuracy validation of rhodomyrtone in plasma were represented by RSD and standard errors of the mean. Five sets of QC samples, comprising low (0.1 μg/mL), medium (50 μg/mL) and high (100 μg/mL) samples, were tested using a single standard curve on the same day for intra-day validation. Inter-day validation was performed by assessing five sets of QC samples on five separate days. By comparing the computed concentrations using a calibration curve based on a known value, the accuracy was ascertained. The peak area ratios of the analytes to IS and the absolute recovery of the rhodomyrtone extracted from the plasma were evaluated. 

The mean recovery was determined at low, medium and high concentrations from five replicates. The long-term storage stability at −80 °C was determined after 15, 30, 60 and 90 d. Bench-top stability testing was also conducted. Samples were spiked in blank plasma and extracted, then analyzed in three replications at 25 °C for 0, 1, 2, 4, 6 and 8 h. The freeze–thaw cycles were stabilized. Rhodomyrtone was spiked in blank plasma, then stored in a freezer (−20 °C) and thawed every 24 h at room temperature (25 °C) for three cycles prior to extraction.

### 3.6. Pharmacokinetic Studies

Male Sprague Dawley rats weighing 250–350 g were housed in separate stainless cages under a controlled environment (temperature at 22 ± 2 °C and relative humidity of 55 ± 10% with 12 h light and 12 h dark cycles) at the Southern Laboratory Animal Facility, Faculty of Science, Prince of Songkla University, Thailand. During the experimental period, they received a standard rodent diet with free access to ad libitum drinking water. Rats were divided into two groups, each consisting of 12 animals. Each group received a single oral dose of rhodomyrtone (dissolved in medium-chain triglyceride) at 50 mg/kg body weight (Group 1) and 100 mg/kg body weight (Group 2). The animals were fasted for 10 h prior to the experiment. An initial blood sample was taken by clipping the end of the animals’ tails. Blood samples (500 μL) were collected at time intervals of 0 (predose), 0.25, 0.5, 1, 2, 3, 4, 5, 6, 7 and 8 h. The samples were transferred to a heparinized microcentrifuge tube and centrifuged at 2500× *g* at 10 °C for 10 min. The plasma samples were stored at −80 °C until HPLC analysis.

### 3.7. Pharmacokinetic Analysis

The individual plasma concentration–time profiles were visually inspected to determine the maximum plasma concentration (C_max_), the minimum plasma concentration (C_min_) and the time to maximum plasma concentration (t_max_). WinNonlin Version 1.1 (Scientific Consulting, Apex, NC, USA) was used to analyze the data and calculate for elimination half-life time (t_1/2_), elimination rate constant (λ_z_), volume of distribution (V_z_/F), area under the concentration–time curve between 0 and 8 h (AUC_0–8_) and the area under the concentration–time curve between 0 and infinity (AUC_0–∞_). The results were expressed as mean values ± standard deviation (SD).

### 3.8. SwissADME Prediction

SwissADME (Swiss Institute of Bioinformatics, Lausanne, Switzerland) software was accessed via http://www.swissadme.ch/ accessed on 30 September 2023 [33]. The molecular structure of rhodomyrtone (Figure 3A) was entered and we performed the prediction. The predicted properties of rhodomyrtone include absorption parameters, pharmacokinetic properties and druglike nature. 

## 4. Conclusions

A simple, cost-effective, reliable and reproducible reverse-phase HPLC-UV method was developed for the routine analysis of rhodomyrtone. The method has been successfully applied for the pharmacokinetics studies of rhodomyrtone in rats. The dose-proportional pharmacokinetic properties of rhodomyrtone release indicate linear pharmacokinetics at a dose range from 50 to 100 mg/kgs. The plasma concentration of the dose range from 50 to 100 mg/kg suggests that rhodomyrtone is suitable for the treatment of bacterial infections of various pathogens with clinical importance. Drug delivery systems such as the inclusion complex and vesicular system could be applied to enhance the bioavailability of rhodomyrtone.

## Figures and Tables

**Figure 1 antibiotics-13-00156-f001:**
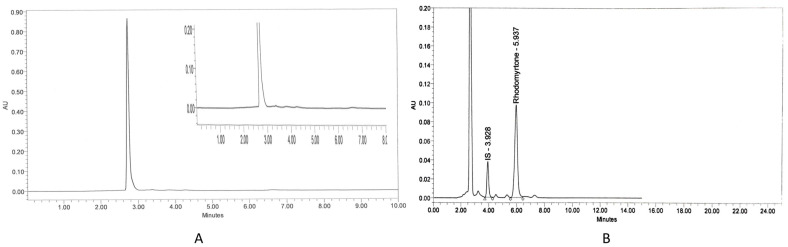
Chromatograms of blank rat plasma (**A**) and rhodomyrtone (16 μg/mL) and internal standard-IS: papaverine (12.5 μg/mL) in rat plasma (**B**).

**Figure 2 antibiotics-13-00156-f002:**
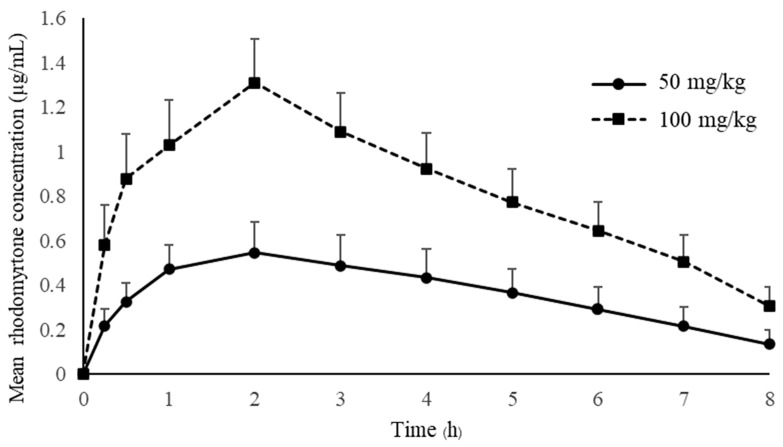
The mean plasma concentration-time of rhodomyrtone after a single oral dose of 50 and 100 mg/kg (mean ± SD).

**Figure 3 antibiotics-13-00156-f003:**
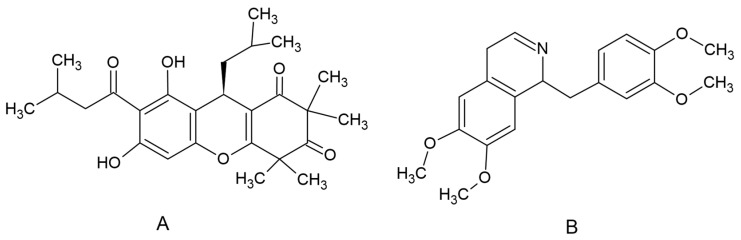
Chemical structure of rhodomyrtone (**A**) and papaverine (**B**).

**Table 1 antibiotics-13-00156-t001:** The lower limit of quantification at 0.04 μg/mL.

LLOQ (μg/mL)	%Accuracy
0.04_1	105.06
0.04_2	108.59
0.04_3	113.83
0.04_4	109.19
0.04_5	95.14
Mean	106.36
SD	0.0028

**Table 2 antibiotics-13-00156-t002:** Intra-day and inter-day precision and accuracy for rhodomyrtone in rat plasma standards.

Rhodomyrtone (µg/mL)	Intra-Day (n = 5)			Inter-Day (n = 5)		
	Mean ± SD	%CV	% Accuracy	Mean ± SD	%CV	% Accuracy
0.1	0.10 ± 0.01	11.83	94.87	0.10 ± 0.01	11.83	94.87
50	50.79 ± 1.52	2.98	101.58	50.79 ± 1.52	2.98	101.58
100	107.46 ± 9.10	8.47	107.46	107.46 ± 9.10	8.47	107.46

**Table 3 antibiotics-13-00156-t003:** Bench-top stability of rhodomyrtone at concentrations 0.1 and 100 µg/mL.

Concentration (µg/mL)	Mean ± SD (n = 5)	CV (%)	Accuracy (%)
0 h			
0.1	0.11 ± 0.007	6.66	108.10
100	97.33 ± 1.83	1.88	97.34
1 h			
0.1	0.09 ± 0.009	9.13	93.38
100	96.58 ± 4.26	4.42	96.58
2 h			
0.1	0.11 ± 0.007	6.36	108.61
100	95.97 ± 4.70	4.90	95.97
4 h			
0.1	0.10 ± 0.005	4.57	101.47
100	94.58 ± 5.22	5.52	94.58
6 h			
0.1	0.10 ± 0.009	8.40	101.24
100	94.58 ± 5.22	2.12	92.47
8 h			
0.1	0.10 ± 0.009	9.56	95.96
100	94.58 ± 5.22	5.67	96.73

**Table 4 antibiotics-13-00156-t004:** Freeze–thaw stability of rhodomyrtone at concentrations of 0.1 and 100 µg/mL.

Concentration (µg/mL)	Mean ± SD (n = 5)	CV (%)	Accuracy (%)
Day 0			
0.1	0.11 ± 0.0026	2.35	110.31
100	103.71 ± 7.44	7.17	103.71
Day 3			
0.1	0.11 ± 0.004	4.057	105.93
100	108.10 ± 2.86	2.645	108.10

**Table 5 antibiotics-13-00156-t005:** Pharmacokinetic parameters of rhodomyrtone dose 50 and 100 mg/kg (n = 12).

Parameters (Unit)	Rhodomyrtone Dose	
	50 mg/kg	100 mg/kg
T_max_ (h)	1.92 ± 0.29	2.00 ± 0.00
C_max_ (μg/mL)	0.57 ± 0.12	1.31 ± 0.20
C_min_ (μg/mL)	0.14 ± 0.06	0.31 ± 0.09
AUC_0–8_ (μg·h/mL)	2.68 ± 1.09	6.65 ± 1.13
λ_z_ (h^−1^)	0.50 ± 0.49	0.30 ± 0.10
AUC_0–∞_ (μg·h/mL)	3.41 ± 1.04	7.82 ± 1.53
V_z_/F (L/kg)	48.47 ± 16.89	46.65 ± 13.82
CL/F (L/kg/h)	15.97 ± 4.69	13.26 ± 2.73
t_1/2_ (h)	2.20 ± 0.74	2.53 ± 0.92

**Table 6 antibiotics-13-00156-t006:** SwissADME predicted the physicochemical properties of rhodomyrtone.

Properties	Predicted Value
Estimated water solubility (mg/mL)	4.21 × 10^−4^
Consensus Log P	4.55
Gastrointestinal absorption	High
Bioavailability Score	0.56

**Table 7 antibiotics-13-00156-t007:** Rhodomyrtone MIC of common pathogens with clinical significance.

Pathogens	MIC (μg/mL)	References
*Staphylococcus epidermidis* ATCC 35984 (biofilm positive)	0.39	[38]
*Bacillus cereus*	0.39	[38]
*Streptococcus mutans*	0.19	[36]
*Staphylococcus aureus* ATCC 29213	0.5	[16,38]
*Staphylococcus aureus* clinical isolated	0.5–2	[39]
MRSA clinical isolated	0.39–0.78	[16,36,38]

## Data Availability

Data are contained within the article.

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
