# Peer review of "A Novel Antibiotic, Rhodomyrtone: Pharmacokinetic Studies in a Murine Model and Optimization and Validation of High-Performance Liquid Chromatographic Method for Plasma Analysis"

_antibiotics, 2024, doi:10.3390/antibiotics13020156_

Round 1

Reviewer 1 Report

Comments and Suggestions for Authors

Manuscript Number: antibiotics-2751148

Manuscript title: A Novel Antibiotic, Rhodomyrtone: Pharmacokinetic Studies in a Murine Model and Optimization and Validation of High Performance Liquid Chromatographic Method for Plasma Analysis

The Reviewer recommends above mentioned paper for publication in Antibiotics after minor revision.

Minor comments

  1. Materials and Methods there are some inaccuracies between this section and Result & Discussion section:

a)       In M&M (Sample preparation subsection) the Authors in lines 242 to 247 described sample preparation which differ from description in lines 90-91. Please check it. 

b)       Also in this same section there are inaccuracies in the description of mobile phase composition 80:20 (line 95) vs 82:18 (line 256).

c)       How many points were used to create calibration curve?

d)       In reviewer’s opinion the calibration curve range was extremely high (3200 times differences between first and last cc point) – is this correct? If yes, why Authors used such “wide” curve if they measured up to 1.5 ug/mL? Such a “wide” curve is less precise.

e)       Also the reviewer can not “recreate” the sample preparation protocol. In what volume the rhodomyrtone was added to the samples under validation protocols? According the description from lines 238-240 the concentration of rhodomyrtone in working standard solution is quite low when we want to create a curve with 128 ug/mL of drug in the last curve point (this volume was probably half of the volume of matrix). Please prepare accurate sample preparation protocol that could be replicable for other researchers (especially what volume of working standard solution of standards was added to what volume of matrix etc.). If this manuscript is in 50% focused on the method, description of the method should be perfect.

  1. Results and Discussion

a)     The Authors some times compare the rhodomyrtone to linezolid and some times to vancomycin. It seems that Authors used comparison to other antimicrobials, only when it is comfortable for them. The comparison is needed, however Authors should create some “reason” of using the other drugs for comparisons. Also the reviewer do not agree that vancomycin is administered only in parenteral route – the weak absorption from the GIT is a huge advantage when this antimicrobial is used against Clostridium difficile (lines 192-193).

b)     There are to many speculations in the manuscript. The results are only from the plasma, and only after oral administration. Because of this Authors should avoid too many speculation:

v  Lines 200 to 205 – we do not know what is the absorption rate, so the Authors should not assumed the distribution of this drug.

v  Also the reviewer do not agree with the Authors that the bioavailabilty will be high (table 6). Based on the concentrations of rhodomyrtone after 100 mg/kg and comparing it to other drugs we can assume that bioavailabilty will be around 10% - this remain intravascular or intraperitoneal drug administration study.

v  What mean SwissADME – how Authors calculate this – please add this to M&M description with the literature 22. Also, the Authors should be very careful using this tool, for preparing the assumptions about PK of drugs.

v  The sentence from line 122-123 is unclear – what Authors wanted to say?

v  Please give the MIC for selected bacteria in the table. Thanks to this the readers do not have to looking for this values in the literature.

v  Sentences from lines 168 to 172 are a little confusing – the efficiency of the  rhodomyrtone is time or concentration dependent?

v  The evidence indicated that the dosages used in this study were adequate for controlling the infections” the reviewer is not sure that it is correct conclusion.

v  Also in the reviewer opinion the sentences “There was no sign of entero-hepatic recirculation after the oral dose within 8 h of the experiment. However, more extended blood samplings and charcoal administration is needed to confirm the entero-hepatic recirculation effect [26].” are redundant.  

 In reviewer opinion this manuscript is written quite well, however there are some inaccuracies. Nevertheless the research are novelty and interesting and should be published in Antibiotics after revision.

Reviewer 2 Report

Comments and Suggestions for Authors

Author has written manuscript well but below are some suggestions that are needed to be addressed before accpetance of manuscript:

An interesting relation for selection of particular HPLC method parameters can be given. Or a short explaination can be given on way to selection of column, mobile phase based on drug properties.

During sample preparation, does 75 ul ACN sufficient for ppt of priteins. injection volume is 80 ul. wasnt it so precautious during collecting supernatent. Even syring filter will absorb some volume of solvent. and organic solvent can dissolve the filter. Do you faced these problems, how you overcome. please explain it in manuscript.

in sample preparation (result section) mobile phase ratio is 80:20, while in materia method section (sample prep) it is 82:18.

ICH Q2R1 guideline is for analytical validation, I believe you are following bioanalytical validation (Please read draft for Bioanalytical validation either by USFDA/ICH). Replace it in Validation of HPLC Method section.

Guidelines have mentioned criteria for selection of LLOQ, please mention that in manuscript. 

According to guidelines, accuracy and precision can be performed at 4 levels (lloq, lqc, mqc and hqc). authors have done it for 3 levels only. please explain or write in manuscript that it should perform at 4 levels.

I have doubt about this sentense "The large volume of distribution 200 indicated that rhodomyrtone was either protein bounded or distributed to the peripheral 201 tissues such as adipose tissue, skeletal muscle, and interstitial fluid" Since your extraction recovery is 100%, can protein binding is repsonsible for high VD. I think Vd is related to distribution and elimination of drug. Please explain . 

Please confirm "Thus, we may classify rhodomyr- 165 tone as a biopharmaceutics classification system (BCS) class II compound" how you classify the drug as BCS class II. BCS biowaiver explain assigning the drug molecule as highly permeable when it shows absorption >85%. Please confirm.  

Bioanlytical validation is not limited to only paramters mentioned in manuscript. Please mention remained paramters that are not performed so reader can understand and implement in their studies.

Reviewer 3 Report

Comments and Suggestions for Authors

This article ‘A Novel Antibiotic, Rhodomyrtone: Pharmacokinetic Studies in a Murine Model and Optimization and Validation of High Performance Liquid Chromatographic Method for Plasma Analysis’ introduces the development and validation of the HPLC methods for analysis the rhodomyrtone in rats’ plasma. This article has certain practical significance and scientific nature. However, there are still some problems that need to be improved.

1.     Regarding the conclusion of linear PK characteristics, the authors only based on the AUC of dose 50 and 100 mg/kg. Please give more data on the lower dose or higher dose before give the linear PK characteristics conclusion.

2.     Moreover, the 302nm was choose as the detection wavelength of HPLC-UV. Please provide appropriate evidence data, such as UV scanning spectra.

3.     In rat plasma pharmacokinetic experiments, it is not enough to have blood sampling points only within 8 hours. Please test the sampling points of 12 hours and 24 hours.

4.     The measurement result of LOQ is 0.04ug/mL. This value is not having enough sensitive and accurate for the determination of rhodomyrtone in rats’ plasma. Please develop a suitable method, with the LOQ lower than 0.02ug/mL, to ensure the measurement requirements at dose 50mg/kg. Moreover, recommend the authors to try using HPLC-MS/MS for method development.

5.     Regarding the determination methods, please discuss the reason why not detect the main metabolites of rhodomyrtone in rats.

6.     Considering the reference significance of the rhodomyrtone PK characteristics, please consider give the data of oral absolute bioavailability.

7.     Regarding the Figure 1: please zoom out the vertical axis of figure1A from 0.00 to 0.20 AU.

Comments on the Quality of English Language

Good.

Reviewer 4 Report

Comments and Suggestions for Authors

The manuscript describes the development and validation of HPLC-UV method for the quantitation of rhodomyrtone and its application to a PK study in rats.

There are 2 main concerns about the study-

1) If 1mg/ml of DMSO stock solution of rhodomyrtone was used, what is the final concentration of DMSO in the calibration standards. If the % DMSO varies among the standards that may affect the solubility of the drug. Also if the % DMSO is significantly high in the standards, it means the standard solvent composition is significantly different from the experimental samples.

2) If 500ul blood is withdrawn at each timepoint, it adds up to be 5.5 ml for 11 timepoints. That is a lot of blood to be withdrwan from rat in such a short duration. Cumulative blood withdrwals should not exceed 2ml for 300 g rat. This high blood volume withdrawal will adversely affect the PK study outcome.

Both these are major concerns for the study and should be addressed.

Round 2

Reviewer 2 Report

Comments and Suggestions for Authors

Authors have adreesed all the queries. I think the manuscript is now suitable for acceptance.

Author Response

We thank the reviewer for their time reviewing and suggesting valuable comments on our manuscript.

Reviewer 3 Report

Comments and Suggestions for Authors

The author failed to provide a convincing response to the reviewer's suggestions. For example: the commons No.1 to 4, and 6 in the first round review were not carefully revised. Please make thoroughly revise on the shortcomings in this manuscript according to scientific requirements.

Comments on the Quality of English Language

Just ok.

Reviewer 4 Report

Comments and Suggestions for Authors

The authors have addressed the comments that were raised, satisfactorily.

Author Response

(The authors gave the same response as above.)
